# An Active Multi-Object Ultrafast Tracking System with CNN-Based Hybrid Object Detection

**DOI:** 10.3390/s23084150

**Published:** 2023-04-21

**Authors:** Qing Li, Shaopeng Hu, Kohei Shimasaki, Idaku Ishii

**Affiliations:** Smart Robotics Laboratory, Graduate School of Advanced Science and Engineering, Hiroshima University, 1-4-1 Kagamiyama, Higashi-Hiroshima 739-8527, Hiroshima, Japan

**Keywords:** high-speed vision, convolutional neural network (CNN), template matching (TM), multi-object tracking

## Abstract

This study proposes a visual tracking system that can detect and track multiple fast-moving appearance-varying targets simultaneously with 500 fps image processing. The system comprises a high-speed camera and a pan-tilt galvanometer system, which can rapidly generate large-scale high-definition images of the wide monitored area. We developed a CNN-based hybrid tracking algorithm that can robustly track multiple high-speed moving objects simultaneously. Experimental results demonstrate that our system can track up to three moving objects with velocities lower than 30 m per second simultaneously within an 8-m range. The effectiveness of our system was demonstrated through several experiments conducted on simultaneous zoom shooting of multiple moving objects (persons and bottles) in a natural outdoor scene. Moreover, our system demonstrates high robustness to target loss and crossing situations.

## 1. Introduction

Multi-target tracking and high-definition image acquisition are important issues in the field of computer vision [1]. High-definition images of many different targets can provide more details, which is helpful for object recognition and improves the accuracy of image analysis. It has been widely used in traffic management. [2], security monitoring [3], intelligent transportation systems [4], robot navigation [5], auto pilot [6], and video surveillance [7].

However, there is a contradiction between wide field of view and high-definition resolution, as shown in Figure 1. The discovery and tracking of multiple targets depends on a wide field of view. While a panoramic camera with a short focal length can provide a wide field of view, the definition of the image is low. A telephoto camera is the exact opposite of a panoramic camera. Using a panoramic camera with a larger resolution is a feasible solution; however, it requires greater expenditure and larger camera size [8]. With the rapid development of deep learning in the image field, the super-resolution reconstruction method based on autoencoding has become the mainstream, and its reconstruction accuracy is significantly better than that of traditional methods [9]. However, due to the huge network model and large amount of model training required in the super-resolution method based on deep learning, there are defects in the reconstruction speed and the flexibility of the model [10].

Therefore, researchers have tried to use telephoto cameras to obtain a larger field of view and track multiple targets. A feasible solution is to stitch the images obtained from a telephoto camera array together into high-resolution images and track multiple targets [11]. Again, this results in greater expenditure and an increase in device size. Another research method is to make the telephoto camera an active system by mounting it on a gimbal. Through the horizontal and vertical movement of the gimbal, the field of view of a pan-tilt-zoom (PTZ) camera can be changed to obtain a wide field of view [12]. However, the original design of such a gimbal camera is not intended for multi-target tracking. Due to the limited movement speed of the gimbal and the size of the telephoto lens, it is difficult for gimbal-based PTZ cameras to move at high speeds and observe multiple objects [13]. Compared to traditional camera systems operating at 30 or 60 fps, high-speed vision systems can work at 1000 fps or more [14]. The high-speed vision system acquires and processes image information with extremely low latency and interacts with the environment through visual feedback control [15]. In recent years, a galvanometer-based reflective camera system has been developed that can switch the perspective of a telephoto camera at hundreds of frames per second [16]. This reflective PTZ camera system is able to virtualize multiple virtual cameras in a time-division multiplexing manner in order to observe multiple objects [17]. Compared with traditional gimbal-based and panoramic cameras, galvanometer-based reflective PTZ cameras have the advantages of low cost, high speed, and high stability [18], and are suitable for multi-target tracking and high-definition capture.

However, the current galvanometer-based PTZ cameras rarely perform active visual control in the process of capturing multiple targets. Instead, they mainly rely on panoramic cameras, laser radars, and photoelectric sensors to obtain the positions of multiple targets, and finally use reflective PTZ cameras for multi-angle capture [19]. Due to the impact of detection delay and accuracy, it is difficult for multiple objects to be tracked smoothly. With the victory of AlexNet in the visual competition, CNN-based detectors continue to develop, and can now detect various objects in an image at a dozens of frames per second [20]. For high-speed vision at a speed of hundreds of frames per second, however, it is difficult to achieve real-time detection with deep learning.

This study aims to utilize a reflective PTZ camera system to track multiple objects and to capture high definition images with low latency. A reflective PTZ camera system switches perspectives to track multiple objects at 500 fps per second by implementing 2-ms-latency visual feedback control. The high-speed vision feedback control relies on CNN-based hybrid detection methods [21]. Compared with the previous system, this work achieves the following: (1) the acquisition of images with large field of view and high resolution, (2) simultaneous observation of up to 20 objects at a speed of 25 fps; and (3) active tracking of multiple fast-moving objects with no-latency detection. The subsequent parts of this study are organized as follows: related works and research are presented in Section 2; a detailed algorithmic analysis and concept illustration are presented in Section 3; Section 4 presents a full description of the experimental test platform, followed by a discussion of the test; finally, our conclusions are presented in Section 5.

## 2. Related Works

This study aims to track multiple tragets by switching the perspectives of PTZ cameras simultaneously. This is closely related to the research on object detection, multiple object tarcking, and high-speed vision. In the next two sections, we provide a brief review of related works.

### 2.1. Object Detection

Object detection is a computer vision task that involves detecting instances of semantic objects of a certain class (such as a person, bicycle, or car) in digital images and videos [22]. The earliest research in the field of object detection can be traced back to the Eigenface method for face detection proposed by researchers at MIT [23]. Over the past few decades, object detection has received great attention and achieved significant progress. Object detection algorithms are roughly divided into two stages, namely, traditional object detection algorithms and the object detection algorithms based on deep learning [24].

Due to the limitations of computing speed, traditional object detection algorithms focus mainly on pixel information in images. Traditional object detection algorithms can be divided into two categories, sliding window-based methods [25] and region proposal-based methods [26]. The sliding window-based approach achieves object detection by sliding windows of different sizes over an image and classifying the contents within the different windows. Many such feature extraction methods have been proposed, such as histogram of oriented gradients (HOG), local binary patterns (LBP), and Haar-based classifiers. These features, in conjunction with traditional machine learning methods such as SVM and BOOST, have been widely applied in pedestrian detection [27] and face recognition [28]. Region proposal-based methods achieve object detection by proposing regions in which objects may exist in generated images, then classifying and regressing these regions [29].

Traditional algorithms have been proven effective; however, with continuous improvements in computing power and dataset availability, object detection technologies based on deep learning have gradually replaced the traditional manual feature extraction methods and become the main research direction. Thanks to continuous development, convolutional neural network (CNN)-based object detection methods have evolved into a series of high-performance structural models such as AlexNet [30], VGG [31], GoogLeNet [32], ResNet [33], ResNeXt [34], CSPNet [35], and EfficientNet [36]. These network models have been widely employed as backbone architectures in various CNN-based object detectors. According to the differences in the detection process, object detection algorithms based on deep learning can be divided into two research directions, One-Stage and Two-Stage [37]. Two-stage object detection algorithms transform the detection problem into a classification problem for generated local region images based on region proposals. Such algorithms generate region proposals in the first stage, then classify and regress the content in the region of interest in the second stage. There are many efficient object detection algorithms that use a two-stage detection process, such as R-CNN [38], SPP-Net [39], Fast R-CNN [40], Faster R-CNN [41], FPN [42], R-FCN [43], and DetectoRS [44]. R-CNN was the earliest method to apply deep learning technology to object detection, reaching an MAP of 58.5% on the VOC2007 data. Subsequently, SPP-Net, Fast R-CNN, and Faster R-CNN sped up the running speed of the algorithm while maintaining the detection accuracy. One-stage object detection algorithms, on the other hand, are based on regression, which converts the object detection task into a regression problem for the entire image [45]. Among the one-stage object detection algorithms, the most famous are single shot multibox detector (SSD) [46], YOLO [47], RetinaNet [48], CenterNet [49], and Transformer [50]-based detectors [51]. YOLO was the earliest one-stage target detection algorithm applied to actual scenes, obtaining stable and high-speed detection results [52]. The YOLO algorithm divides the input image into S×S grids, predicts *B* bounding boxes for each grid, and then predicts the objects in each grid separately. The result of each prediction includes the location, size, confidence of the bounding box, and the probability that the object in the bounding box belongs to each category. This method of dividing the grid avoids a large number of repeated calculations, helping the YOLO algorithm to achieve a faster detection speed. In follow-up studies, algorithms such as YOLOv2 [53], YOLOv3 [54], YOLOv4 [55], YOLOv5 [56], and YOLOv6 [57] have been proposed. Owing to its high stability and detection speed, in this study we use yolov4 as the AI detector.

### 2.2. Multiple Object Tracking (Mot)

Multiple object tracking detects and tracks multiple targets in videos, such as pedestrians, vehicles, and animals. It is an important research direction in the field of computer vision, and has been widely applied in intelligent surveillance and behavior recognition [58]. Early classical tracking methods, such as Meanshift [59], particle filtering [60], KCF [61], and MOSSE [62], mainly focused on single-object tracking. With the rapid development of CNNs, detection-based multi-object tracking methods have quickly become the mainstream research direction.

Currently, there are three popular research directions in multi-object tracking: detection-based MOT, detection and tracking-based joint MOT, and attention-based MOT. In detection-based MOT algorithms, object detection is performed on each frame to obtain image patches of all detected objects. A similarity matrix is then constructed based on the IoU and appearance between all objects across adjacent frames, and the best matching result is obtained using a Hungarian or greedy algorithm; representative algorithms include SORT [63] and DeepSORT [64]. In detection and tracking-based joint MOT algorithms, detection and tracking are integrated into a single process. Based on CNN detection, multiple targets are fed into the feature extraction network to extract features and directly output the tracking results for the current frame. Representative algorithms include JDE [65], MOTDT [66], Tracktor++ [67], and FFT [68]. The attention mechanism-based MOT is inspired by the powerful processing ability of the Transformer model in natural language processing. Representative algorithms include TransTrack [69] and TrackFormer [70]. TransTrack takes the feature map of the current frame as the key and the target feature query from the previous frame and a set of learned target feature queries from the current frame as the input query of the whole network.

### 2.3. High-Speed Vision

High-speed vision is a computer vision technology that aims to realize real-time image recognition and analysis through the use of fast and efficient image processing algorithms at high frame rates of 1000 fps or more. It is an important direction in the field of computer vision and is used in a variety of applications, such as intelligent transportation [71], security monitoring [72], and industrial automation [73].

High-speed vision has two properties: (1) the image displacement from frame to frame is small, and (2) the time interval between frames is extremely short. In order to realize vision-based high-speed feedback control, it is necessary to process massive images in a short time. Unfortunately, current image processing algorithms, such as noise reduction, tracking, and recognition, are all based on traditional image data involving dozens frames per second. An important idea in high-speed image processing is that it reduces the amount of work required for small-scale shifts between high-speed frames. Field programmable gate arrays (FPGAs) and graphics processing units (GPU), which support massively parallel operations, are ideal for processing two-dimensional data such as images. In [74], the authors presented a high-speed vision platform called $H^{3}$ vision. This platform employs dedicated FPGA hardware to implement image processing algorithms and enables simultaneous processing of a 256×256 pixel image at 10,000 fps. The hardware implementation of image processing algorithms on an FPGA board provides high performance and low latency, making it suitable for real-time applications that require high-speed image processing. In [75], a super-high-speed vision platform (HSVP) was introduced that was capable of processing 12-bit 1024×1024 grayscale images at a speed of 12,500 frames per second using an FPGA platform. While the fast computing speed of FPGA is ideal for high-speed image processing, its programming complexity and limited memory capacity can pose significant challenges. Compared with FPGA, GPU platforms can realize high-frame-rate image processing with lower programming difficulty. A GPU-based real-time full-pixel optical flow analysis method was proposed in [76].

In addition to high-speed image processing, high-speed vision feedback control is very important. Gimbal-based camera systems are specifically designed for image streams with dozens of frames per second. Due to the limited size and movement speed of the camera, it is often difficult to track objects at high speeds while simultaneously observing multiple objects. Recently, a high-speed galvanometer-based reflective PTZ camera system was proposed in [77]. The PTZ camera system can acquire images from multiple angles in an extremely short time and virtualize multiple cameras from a large number of acquired image streams. In [78], a novel dual-camera system was proposed which is capable of simultaneously capturing zoomed-in images using an ultrafast pan-tilt camera and wide-view images using a separate camera. The proposed system provides a unique capability for capturing both wide-field and detailed views of a scene simultaneously. To enable the tracking of specific objects in complex backgrounds, a hybrid tracking method that combines convolutional neural networks (CNN) and traditional object tracking techniques was proposed in [21]. This hybrid method achieves high-speed tracking performance of up to 500 fps and has shown promising results in various applications, such as robotics and surveillance.

## 3. Active Multi-Object Ultrafast Tracking with CNN-Based Hybrid Object Detection

We propose the concept of wide field of view registration and high-speed multi-object active tracking by virtual cameras using a galvanometer-based reflective PTZ camera, as shown in Figure 2. This high-speed reflective PTZ camera can change the view thousands of times per second to scan the monitoring area. Objects detected during scanning are registered as tracking targets, then the reflective PTZ camera system switches perspectives between different objects at an ultrafast speed for tracking. By classifying and combining frames of different views, multiple virtual cameras with a frame rate of hundreds of frames can be formed. As mentioned in Section 1, current CNN-based object detectors often have dozens of milliseconds of latency from input frames to output results. Compared to ultra-high-speed galvano-mirror control, detection latencies can negatively impact vision-based feedback control, causing skipped frames without object detection. To address this issue, we propose a framework called HFR multiple target tracking, which combines high-speed TM-based trackers with CNN-based object detectors to achieve low-latency visual feedback at hundreds of Hz. This framework enables the tracking of multiple fast-moving objects, and can be used in real-time applications such as robotics, surveillance, and autonomous navigation.

As shown in Figure 3, the whole multi-object tracking process mainly includes two processes, namely, the new object registration process and the multi-object tracking process. The object registration process first scans the surveillance area at an ultrafast speed and stitches together a high-resolution panoramic image. Subsequently, the CNN-based detector detects the image frame-by-frame, looking for objects of interest to complete the registration. After object registration process, the multi-target tracking process changes the field of view to observe different objects at an extremely fast speed. Meanwhile, we use a CNN-based hybrid detection method in each virtual camera for low-latency visual feedback control. If the object is lost, the object registration process is restarted to register new objects.

### 3.1. New Object Registration Process

The high-speed reflective PTZ camera system captures frames of different angles through ultra-fast rotating two-axis galvano-mirrors. During the scanning process of the monitoring area, all captured high-speed frames, denoted as $I_{h}\left(t\right)$, and control angles, denoted as $v\left(t\right)=\left\{u_{pan}\left(t\right),u_{tilt}\left(t\right)\right\}$, are stored in the frame set *F* and angle set *V*. Meanwhile, the CNN-based detector performs object detection on the frame set *F* during the scanning process. The detected objects in an input frame $I_{h}\left(t\right)$ at time *t* are expressed as follows:(1)$$D\left(I_{h}\left(t\right)\right)=\left\{d^{1}\left(t\right),d^{2}\left(t\right),\ldots ,d^{j}\left(t\right),\ldots ,d^{J}\left(t\right)\right\},$$
where *D* denotes an operator of the CNN-based object detection. For the *j*-th detected object (j=1,⋯,J), each detection result $d^{j}\left(t\right)$ is composed of the following parameters:(2)$$d^{j}\left(t\right)=\left\{o_{x}^{j}\left(t\right),o_{y}^{j}\left(t\right),w^{j}\left(t\right),h^{j}\left(t\right),p^{j}\left(t\right),c^{j}\left(t\right)\right\},$$
where $o^{j}\left(t\right)=\left(o_{x}^{j}\left(t\right),o_{y}^{j}\left(t\right),w^{j}\left(t\right){,h}^{j}\left(t\right)\right)$ denotes the bounding box of *j*-th detection object. In addition, $p^{j}\left(t\right)$ and $c^{j}\left(t\right)$ denote its detection confidence and object class, respectively. In this article, the detection algorithm used in AI detection is YOLOv4, which is currently a very mature detection algorithm with low latency and stable detection time. If an object is detected, we obtain the control angle $v_{i}$ of the *i*-th object as follows: (3)$$v_{i}=\left\{u_{pan}^{i}\left(t\right)+\varepsilon_{pan}d_{pan},u_{tilt}^{i}\left(t\right)+\varepsilon_{tilt}d_{tilt}\right\},$$
where $u_{pan}^{i}\left(t\right)$ and $u_{tilt}^{i}\left(t\right)$ denote the control angle of the pan and tilt when the frame is captured at time *t*, $\varepsilon_{pan}$ and $\varepsilon_{tilt}$ denote the pixel deviation between the center of the object and the frame center, and $d_{pan}$ and $d_{tilt}$ denote the gain between the pixel and the angle. Then, the detected object info $r_{i}=\left\{v_{i},c_{i}\right\}$ is registered to the target set $R=\left\{r_{1},r_{2},\cdots ,r_{n}\right\}$ for high-speed multi-object tracking, while $c_{i}$ denotes the label of the *i*-th object.

### 3.2. Multi-Target Tracking Process

The multi-object tracking process is initiated when the target set size *R* is greater than 0. In the image acquisition process of the galvanometer-based reflective PTZ camera (as depicted in Figure 4a, the galvano-mirror movement and camera exposure represent the two primary stages. To cope with high-speed image streams of hundreds of frames per second, we parallelize the image acquisition and processing process and divide the stream into multiple virtual cameras.

YOLOv4 can process only about 30 frames per second, and struggles to keep up with the real-time processing demands of the hundreds of frames per second in each virtual camera. To overcome this challenge, we developed a hybrid algorithm that combines template matching and CNN-based object detection. Specifically, the object template image obtained from the CNN detector is matched with the current image through template matching to update the object position in each virtual camera. Our hybrid algorithm comprises two modes, as illustrated in Figure 4b: (1) playback mode, which performs real-time playback tracking in all intermediate images from the detected frame to the current frame once CNN obtains the object position, and (2) frame-by-frame forward tracking, which matches the template obtained from CNN frame-by-frame with the new input image.

To reduce the processing delay caused by the CNN, we activate the instant playback tracking mode to address the deviation caused by the difference between a newly detected fast-moving object’s position and its position in the current frame. After updating the template of the TM-based tracker with the detected object area, the TM-based tracker estimates the target position in the current image and the new object position in all frames from the detected input frame to the current frame. The estimated target area of the current frame during playback tracking is used to determine the template of the TM-based tracker and the initial position of frame-by-frame tracking.

Below, we describe the algorithm used in the hybridized object-tracking approach. We denote the time intervals of the input HFR images and CNN-based object detection as $\tau_{h}$ and $\tau_{d}$, respectively, with $\tau_{d}$ being much larger than $\tau_{h}$ and equal to $m\tau_{h}$.

(1) Updating object template using CNN

Due to detection latency, CNN detectors skip frames and continuously detect images from the *n*th virtual camera. The objects detected in an input image $I_{n}\left(t_{d}\right)$ from the *n*th virtual camera at time $t_{d}$ can be described as follows:(4)$$D\left(I_{n}\left(t_{d}\right)\right)=\left\{d_{n}^{1}\left(t_{d}\right),d_{n}^{2}\left(t_{d}\right),\ldots ,d_{n}^{J}\left(t_{d}\right)\right\}.$$

The definition of the detection results $d_{n}^{j}\left(t_{d}\right)$ is the same as that in Equation (Equation 2). Results that differ from the target labels $c_{n}$ tracked by the *n*th virtual camera are initially eliminated. To update the template $I_{n}$ of the *n*th virtual camera for the *S* detection results for which the class is the same as $c_{n}$, we use the following method:(5)$$T_{n}=\left\{\begin{array}{c} \underset{T_{s}}{\mathrm{argmax}}NCC\left(T_{s},T_{n}^{\prime }\right)\left(S>0\right)\\ don^{\prime }t\;update,\left(S=0\right), \end{array}\right.$$
(6)$$NCC\left(T_{s},T_{n}^{\prime }\right)=\frac{\mathrm{Cov}(T_{s},T_{n}^{\prime })}{\sqrt{\mathrm{Var}\left(T_{s}\right)\mathrm{Var}\left(T_{n}^{\prime }\right)}}.$$

Here, $T_{n}$ and $T_{n}^{{}^{\prime }}$ are the current and last template of the *n*th virtual camera, respectively. The NCC [79] (Normalized Cross Correlation) algorithm is used to measure the similarity of templates.

(2) TM-Based HFR Tracking

(a) Frame-by-frame forward tracking: when there is no template update the position $p(t_{n})$ of the tracked target in the image is obtained directly through the SDS (standard deviation of squares) equation, as shown below:(7)$$p\left(t_{n}\right)=p^{\prime }\left(t_{n}\right)+\underset{\left|x\right|\leq Ran,\left|y\right|\leq Ran}{\arg\min}E\left(x,y\right),$$
(8)$$E\left(x,y\right)=\frac{\sum_{x^{\prime },y^{\prime }}{\left(T_{n}\left(x^{\prime },y^{\prime }\right)-I_{n}\left(x_{n}^{\prime }+x+x^{\prime },y_{n}^{\prime }+y+y^{\prime }\right)\right)}^{2}}{\sqrt{\sum_{x^{\prime },y^{\prime }}T_{n}{\left(x^{\prime },y^{\prime }\right)}^{2}\cdot \sum_{x^{\prime },y^{\prime }}I_{n}{\left(x_{n}^{\prime }+x+x^{\prime },y_{n}^{\prime }+y+y^{\prime }\right)}^{2}}}.$$

Here, $p^{\prime }\left(t_{n}\right)$ is the position of the object in the last image. Due to the advantage of high-speed vision, objects move more slowly between high-speed image sequences and there is less displacement between frames. Accordingly, we can set the search range Ran for object detection in the image to a few pixels. Here, $\left(x_{n}^{\prime },y_{n}^{\prime }\right)$ represents the top-left coordinate of the target region in the image from the previous time step in the *n*th virtual camera, while $I_{n}$ represents the new image from the *n*th virtual camera. Because the algorithm is applied to a real-time high-speed system, we prioritize speed over accuracy and robustness.

(b) Playback tracking during template updating: there is a fatal problem with frame-by-frame template matching, which is that the appearance of a moving object often changes. As time progresses, the tracking becomes unstable. Therefore, when updating the template it is necessary to perform a playback operation. The playback operation refers to the process of performing a sub-forward TM through all image sequences from the time $t_{n}^{\prime }=t_{n}-t_{d}$ at which the previous input image was passed to the CNN until the current time $t_{n}$.
(9)$$p\left(t_{n}^{\prime }+\left(k+1\right)\tau_{n}\right)=p\left(t_{n}^{\prime }+k\tau_{n}\right)+\underset{\left|x\right|\leq Ran,\left|y\right|\leq Ran}{\arg\min}E\left(x,y\right),\left(0<=k\tau_{n}<=t_{d}\right).$$

Here, $\tau_{n}$ represents the time interval between frames for the *n*th virtual camera. As shown in Figure 4b, we perform a replay operation every time we update the template to avoid the problem of changes in the object’s appearance. To reduce latency in CNN-based object detection at $t_{d}$ intervals, playback tracking functions are utilized as delay compensators, while frame-by-frame forward tracking functions serve as frame interpolators, converting target positions from $t_{d}$ intervals to $\tau_{n}$ intervals that match the HFR input images.

## 4. Experiments

### 4.1. System Configuration

To enable tracking of multiple fast-moving targets distributed across a wide area, we developed a high-speed pan-tilt camera system that utilizes an ultrafast galvanometer mirror. The system is capable of tracking multiple moving objects simultaneously with a frame rate of 500 fps. The system includes a high-speed CMOS camera head from Image Source, Bremen, Germany (DFK37BUX287), a two-axis pan-tilt Galvano-mirror from Cambridge Technology, Kansas City, MO, USA (6210H), and a control computer with an Intel i9-9900K processor (3.6 GHz), 64-GB DDR4 RAM, and Windows 10 Home (64-bit). Control signals are sent to the Galvano-mirror via a D/A board (PEX-340416) from Interface Corporation, Hiroshima, Japan.

In this paragraph, we describe the technical specifications of the galvanometer-based reflective PTZ camera system. The camera head has a 50 mm telephoto lens and a color CMOS sensor measuring 720 × 540 pixels. The sensor has a size of 4.96 × 3.72 mm and a pixel size of 6.9 × 6.9 μm. It can capture 8-bit RGB 720 × 540 images at 539 fps and transfer them to a PC via USB 3.1. The galvanometer mirror provides two degrees of freedom gaze control, with a range of −20 to 20 degrees for pan and −10 to 10 degrees for tilt. The mirror can be controlled within 2 ms in the ten-degree range. The the overview and geometry of the galvanometer-based reflective PTZ camera system are presented in Figure 5. Real-time control signals from the computer to the galvano-mirror via the D/A board enable the system to zoom in on and track multiple objects.

### 4.2. Execution Times of Visual Tracking Algorithm

The system captures 640 × 480 images at a rate of 500 fps ($\tau_{h}$ = 2 ms), with the initial search areas determined by the position of the moving object obtained by the new object registration process.

Using YOLOv4 for CNN detection, we can execute object detection with a delay of about $\tau_{d}=33$(m=16) ms using multiple high-speed PTZ virtual cameras. In the implemented YOLOv4, eighty object categories (car, bicycle, sports ball, apple, mouse, etc.) were pre-trained using the COCO dataset, with color images resized to 416 × 416 for the purpose of estimating object regions and labels. We fine-tuned the YOLOv4 pre-trained model to incorporate facial detection, allowing the network to detect human faces. The average latency of the processing pipeline for object detection in our system is $\tau_{l}=30$ ms (L=15). Considering that the frame rate of a high-speed virtual PTZ camera increases with the number of objects to be tracked, the frame rate drops by almost 100 frames. Thus, the displacement of the object between adjacent frames in a virtual PTZ camera is slightly larger than the continuous image stream of 500 fps. The search range in TM-based tracking is set to the 5 × 5 neighborhood (R=4) in both instant playback tracking and forward tracking modes. The sub-images obtained from the template image are adaptively down-sampled according to image size before completing the template matching, thereby speeding up the playback tracking process and keeping the time within 2 ms. Our algorithm enables pan-tilt tracking with 500-fps visual feedback control to track multiple moving targets at the image center $\left(c_{x},c_{y}\right)=\left(320,240\right)$. We evaluated the execution times of our algorithm with template sizes of 64 × 64, 128 × 128, 256 × 256, and 512 × 512 pixels and compared the results with those of the following single-object tracking algorithms prepared as tracking APIs in OpenCV 4.5: MIL, BOOSTING, Median Flow, TLD, KCF, GOTURN pre-trained on the ALOV300++ dataset, and MOSSE. Table 1 summarizes the execution times for implementation on 640 × 480 input images using the same PC used for our proposed system. For our algorithm with R=4 and L=6, we show the execution time for (i) playback tracking with template updating, (ii) frame-by-frame forward tracking, and (iii) YOLOv4; YOLOv4 is executed in parallel with the template matching track, and the largest processing delay occurs in playback tracking. Thus, it is necessary to increase the robustness of object tracking under large processing delays. Our algorithm can deliver target positions in real-time, achieving a speed of hundreds of fps or higher through parallel execution with YOLOv4. Compared with other single-object tracking approaches based on either online adaptive templates or pre-trained deep neural networks, our algorithm offers an advantage in terms of processing speed.

### 4.3. Simultaneous Tracking of Twenty Different Objects

We first tested the proposed multi-object tracking system based on reflective mirrors for tracking a large number of targets. Figure 6a shows the overview of the experimental scene. We placed twenty different types of targets, such as cars, bicycles, clocks, and sports balls, on the wall and whiteboard approximately 6 m away from the multi-object tracking system. Among these, the whiteboard was movable and the objects on the whiteboard were able to move along with the motion of the whiteboard. During the object registration process, the galvanomirror-based reflective camera system scanned the monitoring area at a speed of 500 fps. Figure 6b shows the panoramic image stitched together from the reflective camera system following completion of the object registration process. The resulting twenty detected objects were mapped onto the panoramic image in the form of overlays, and their positions in the panoramic image were updated in real-time. The virtual camera labels are shown in “v-cam:n”, where different virtual cameras are assigned different numbers. The virtual camera responsible for scanning is displayed as a green bounding box in the top right corner of the image. If object tracking failed, the virtual camera responsible for scanning was reactivated to search for new objects. Figure 7 presents high-definition images of all tracked objects continuously updated within the image frames at 30 fps. Figure 8 shows the pan and tilt angles of the galvanometer-based reflective PTZ camera when scanning and tracking twenty different targets. The other objects on the wall remained stationary while the objects (the bicycle and car) attached to the whiteboard (virtual cameras 8, 13, 18) moved left and right along with the whiteboard. Because high-speed resources are divided equally, the frame rate of each target is low and certain fast-moving objects cannot be tracked accurately due to their high motion speed.

### 4.4. Low-Latency Pan-Tilt Tracking of Multiple Moving Bottles

Next, we verified the multi-object visual tracking performance of our proposed system at 500 fps. In this experiment, we employed a visual search to automatically detect bottles that were distributed in the surveillance area. By changing the viewpoint, we were able to track multiple bottles at the image centre of a 640 × 480 input image. The experimental environment is depicted in Figure 9. Three bottles were strategically positioned at a distance of about 8 m from the PTZ camera system. Subsequently, two of the bottles were released from a height of approximately 1.7 m while the camera system finished searching and needed to track multiple bottles. A digital camera (model DSC-RX10M3, focal length 30 mm) was positioned adjacent to the camera system to capture 1920 × 1080 images of the surveillance area at 60 frames per second.

The initial stage of the experiment involved a zigzag scan of the monitoring area using the PTZ camera system. Specifically, the pan mirror was adjusted by 2.54 degrees and the tilt mirror by 1.9 degrees for each scan. The final observation range of the pan mirror was set between −17.78 degrees and 17.78 degrees, while the tilt mirror was set to observe within a range of −9.5 degrees to 9.5 degrees. Figure 10 shows the 1920 × 1080 input images from digital camera and panoramic stitched 9600 × 5280 images from the PTZ camera. Our PTZ camera system can obtain 24× higher-definition images of the surveillance area at 3 fps compared with digital cameras. Subsequently, we utilized YOLOv4 to analyze and detect objects in the 165 images captured during the zigzag scan process. Following the completion of object registration, we started a low-latency tracking process to track the detected bottles in real time. Figure 11 shows the 145 × 108 ROI images around the targets from the digital wide-view camera and 640 × 480 input images from the virtual PTZ cameras at t = 19.1 s. In the image obtained from PTZ camera, the characters on the bottles can be clearly read while being robustly tracked. In contrast, only the approximate outline and color of the bottles can be seen in the digital camera image. During tracking, we first picked up two bottles from the table, then released the bottles into a free fall at t = 19.1 s.

Figure 12 shows the pan and tilt angles of the galvanometer-based reflective PTZ camera when scanning and tracking multiple bottles. The system undergoes object registration from 0 to 9 s, after which it transitions to the multi-target tracking process. We simultaneously released two water bottles from a height of about 1.7 m at 19.1 s. The two bottles experienced about 0.6 s of freefall. Figure 13 illustrates the *x* and *y* coordinates of the centroids for the regions of interest (ROIs) that were tracked in the input images during the period t = 9–23 s. Except in the process of falling, the deviation from the center of the image gradually increases, and is otherwise very close to the center of the image (320 × 240).

Figure 14 depicts the tracking status during free fall of bottle 1 when tracking three bottles simultaneously. Figure 14a,b shows the tracking situations based on the CNN hybrid algorithm and YOLOv4, respectively. When using only YOLO tracking, objects leave the field of view quickly at t = 0.3 s as their speed increases. Nevertheless, the CNN-based hybrid tracking algorithm exhibits superior performance in tracking the falling bottle. The velocity of the free-falling bottle is directly proportional to the time it takes to fall. Figure 15 shows the relationship between the velocity and distance from the detection ROI to the image center during free-fall bottle tracking. YOLOv4 tracking is limited in its ability to track objects with a speed greater than 3 m per second. Using our CNN-based hybrid algorithm for tracking, a moving object with a speed of 5.5 m/s is located approximately 45 pixels away from the image center. In theory, it is possible to track three objects moving at a speed of 30 m/s and maintaining a distance of 8 m simultaneously. Finally, we conducted experiments to track three free-falling bottles simultaneously using different tracking methods. Figure 16 shows pixel deviation values between the object position calculated by different algorithms and the object’s actual position during the falling process. The actual position of the object was obtained from offline videos recorded at 30 fps during online system operation. The deviation values shown in the figure represent the specific error at each time step. After being dropped from a height of approximately 1.7 m, the water bottle impacted the ground after approximately 0.6 s. The Boosting, TLD, KCF, and Medianflow tracking methods lost the target within 0.2 s, 0.3 s, 0.4 s, and 0.5 s, respectively. The GOTURN and MOSSE tracking methods, along with our proposed method, were able to maintain tracking until the end of the tracking task. Among these methods, the deviation of the tracked object’s real position is smaller when using our proposed hybrid tracking method based on CNN.

### 4.5. Multi-Person Pan-Tilt Tracking in Wide-Area Surveillance

Subsequently, we designed a scenario more commonly encountered in reality involving multiple individuals in motion. Figure 17a shows the experimental environment captured by the digital camera (focal length = 40 mm). Five individuals were positioned approximately 8 m away from the PTZ camera system. Prior to the completion of the scans and detection, all participants stood still. Figure 17b depicts the scanning and detection outcome at the start of the experiment. Figure 18 depicts the pan and tilt angles of the galvanometer during the experiment as it scanned and tracked different individuals. The upper right corner shows a thread for detecting lost targets and initiating rediscovery. The tracking of multiple individuals commenced at 8.5 s. Between 10 and 13 s, all individuals performed vertical jumps, while from 13 to 15 s, they swayed their bodies horizontally. A loss waiting time of 300 frames was established for each individual. After a waiting time of 300 frames, person 1 was lost at 16.5 s and the scanning thread was restarted; person 1 was eventually rediscovered at 21 s. Two crossings occurred between person 2 and person 3 during 23 to 27 s and 30.5 to 34 s. Sface [80] was employed to extract facial features and conduct similarity matching. In this study, we only utilized facial features for facial discrimination, and did not assign individual IDs for recognition of individuals. During the cross-tracking process, a virtual camera always followed a single person. As shown in Figure 19, person 3 was briefly occluded by person 2 during the initial cross-tracking process and was subsequently re-identified and tracked. In the second crossing process, person 3 was not identified during the short occlusion; thus, the scanning thread had to be restarted, leading to the rediscovery of person 3 at 33 s, as shown in Figure 20. Due to lighting conditions, person 5 was not detected at 26 s, prompting the scanning thread to remain active from that point forward in an attempt to locate person 5. The *x* and *y* coordinate values of the image centroids of the tracked ROIs are depicted in Figure 21. These results demonstrate that our PTZ camera system can track multiple fast-moving objects simultaneously at a high speed. Moreover, the system exhibits high robustness to object loss and occlusion.

## 5. Conclusions

In this study, we developed a multi-object visual surveillance system with 500-fps image processing capabilities able to robustly track multiple objects within a wide area. The effectiveness of our system was demonstrated through two experiments: (1) tracking of multiple free-falling water bottles, and (2) tracking of multiple freely moving individuals. Our system offers three key advantages: (1) rapid generation of high-definition wide-view images; (2) the ability to track up to twenty low-speed moving targets at a maximum rate of 25 fps; and (3) simultaneous tracking of multiple high-speed moving targets with high robustness against object occlusion and loss.

However, there are several limitations faced by the current system; for example, the object registration process in the early stage requires several seconds, during which time the object may continue to move, resulting in target loss during the tracking process. In future work, we plan to incorporate a panoramic camera for pre-detection of interesting targets within the monitoring area.

## Figures and Tables

**Figure 1 sensors-23-04150-f001:**
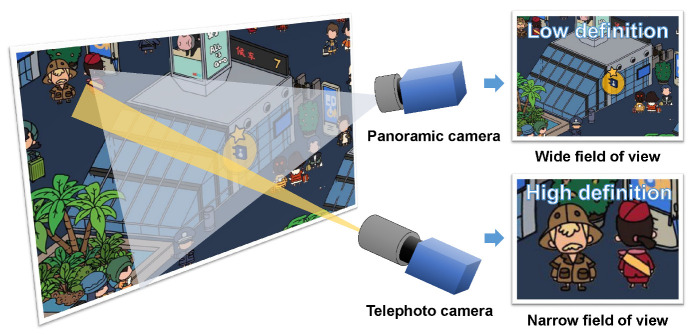
Contradiction between wide field of view and high-definition images.

**Figure 2 sensors-23-04150-f002:**
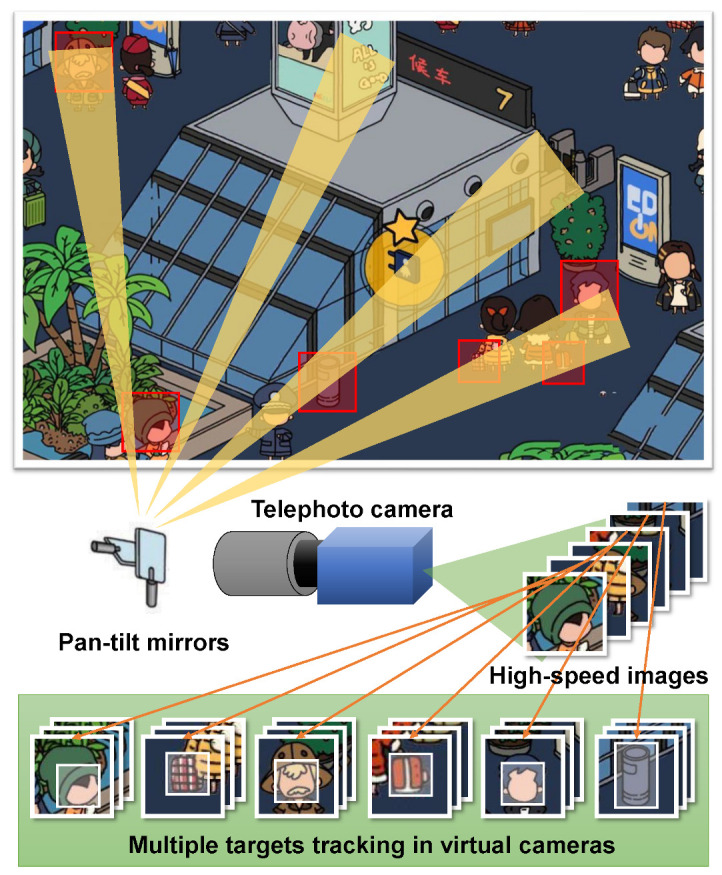
Wide field of view registration and multi-object tracking by virtual cameras using a galvanometer-based reflective PTZ camera.

**Figure 3 sensors-23-04150-f003:**
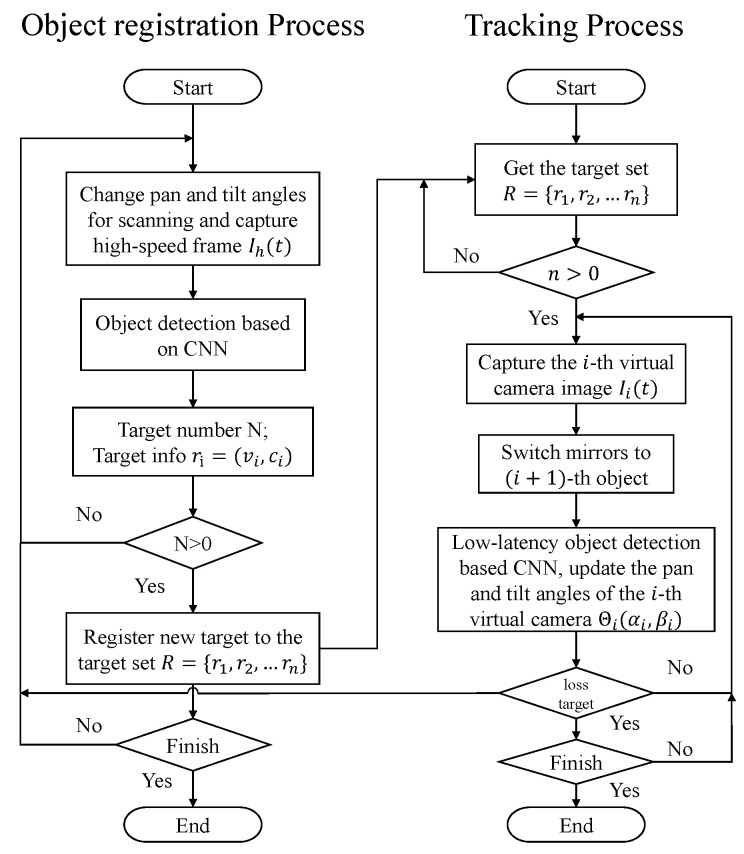
Flowchart of object registration process and multi-object tracking process.

**Figure 4 sensors-23-04150-f004:**
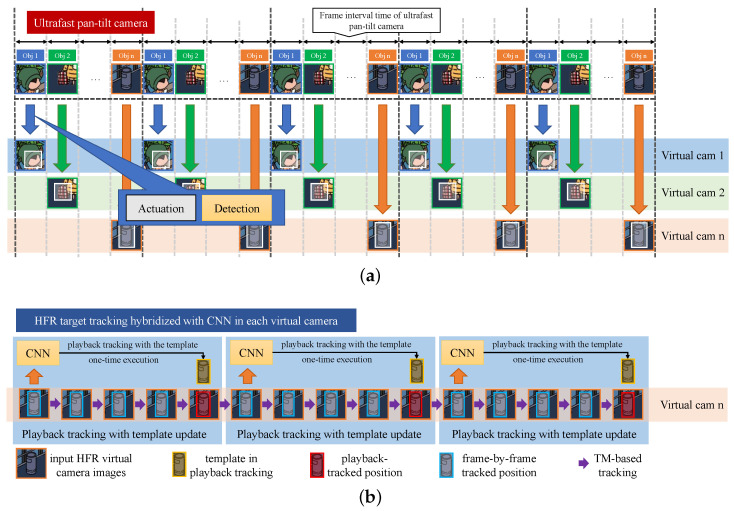
Time-division threaded gaze control process for multiple target tracking based on HFR object detection hybridized with CNN: (**a**) time-division threaded gaze control process for simultaneous multi-object observation and (**b**) HFR target tracking hybridized with CNN in each virtual camera.

**Figure 5 sensors-23-04150-f005:**
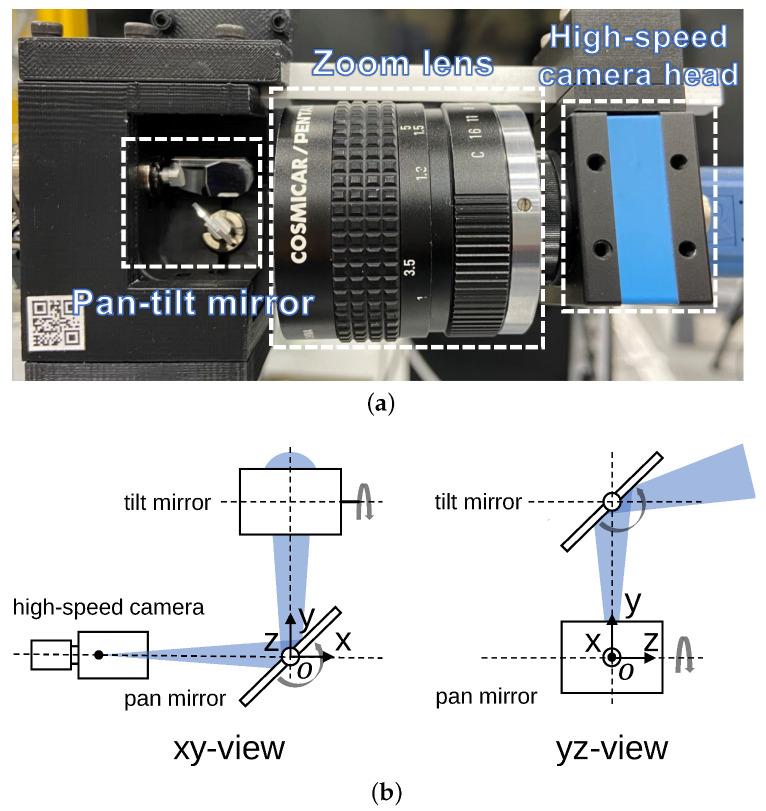
(**a**) Overview and (**b**) geometry of the galvanometer-based reflective PTZ camera.

**Figure 6 sensors-23-04150-f006:**
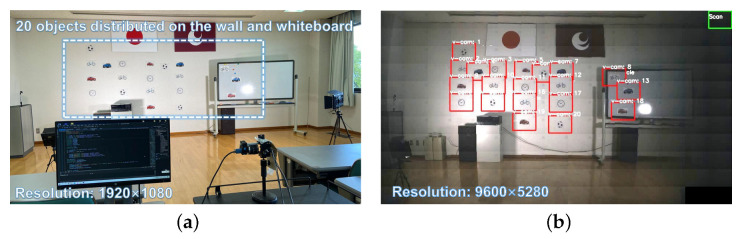
The 1920 × 1080 input images from the digital camera and the panoramic stitched 9600 × 5280 images from the PTZ camera: (**a**) overview of the experimental scene and (**b**) panoramic stitched image from the PTZ camera (targets are pasted on the panoramic image in the form of a red frame texture).

**Figure 7 sensors-23-04150-f007:**
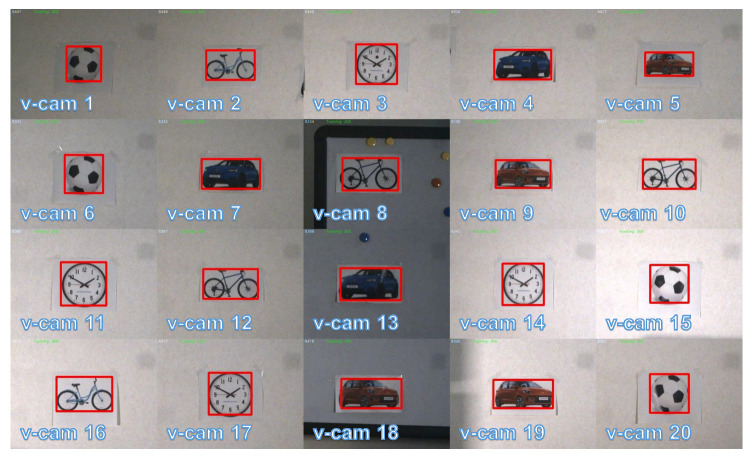
HD images of twenty objects tracked simultaneously.

**Figure 8 sensors-23-04150-f008:**
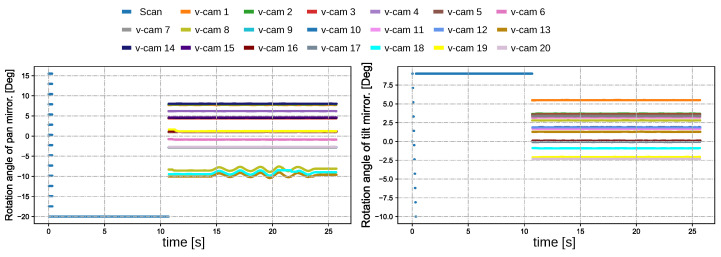
Pan and tilt angles of the galvanometer-based reflective PTZ camera when scanning and tracking twenty different targets.

**Figure 9 sensors-23-04150-f009:**
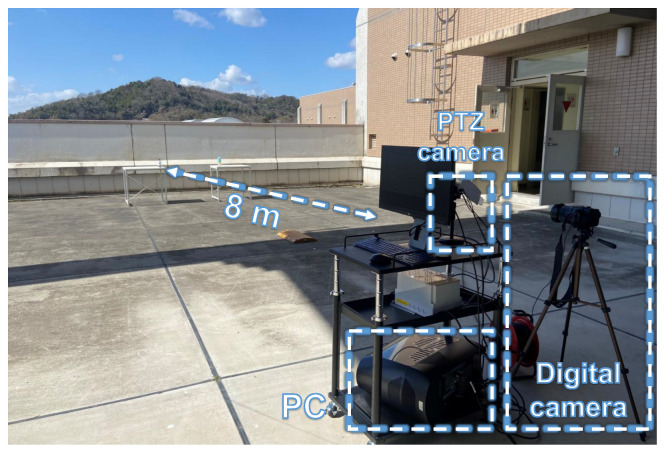
Experimental environment used for tracking multiple moving objects in an outdoor scene.

**Figure 10 sensors-23-04150-f010:**
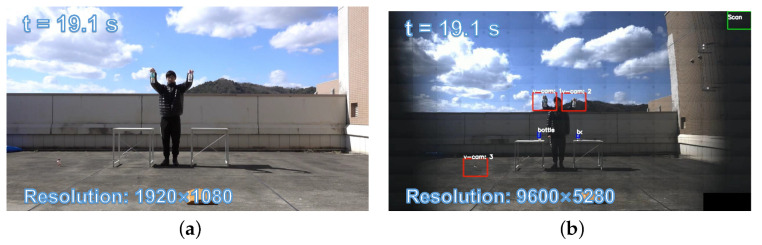
The 1920 × 1080 input images from the digital camera and panoramic stitched 9600 × 5280 images from the PTZ camera: (**a**) input image of digital camera and (**b**) panoramic stitched image from the PTZ camera.

**Figure 11 sensors-23-04150-f011:**
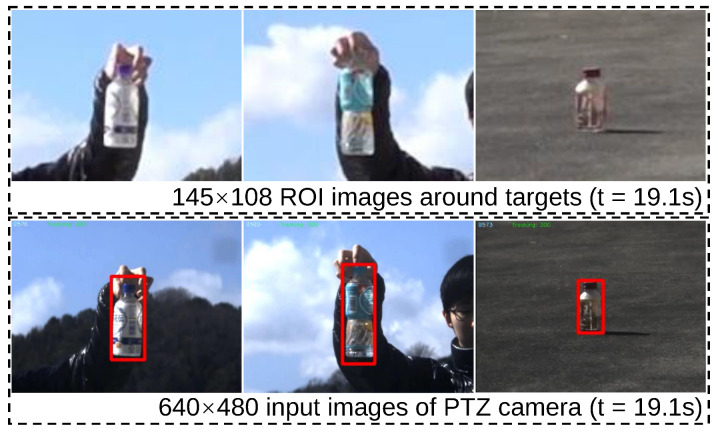
The 145 × 108 ROI images around targets from the digital wide-view camera and 640 × 480 input images from the virtual PTZ cameras (red boxs are the test results).

**Figure 12 sensors-23-04150-f012:**
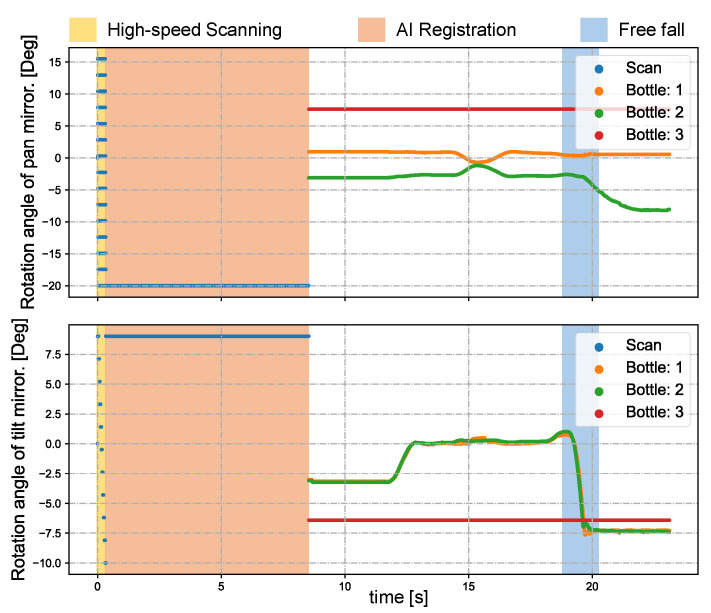
Pan and tilt angles of the galvanometer-based reflective PTZ camera when scanning and tracking multiple bottles.

**Figure 13 sensors-23-04150-f013:**
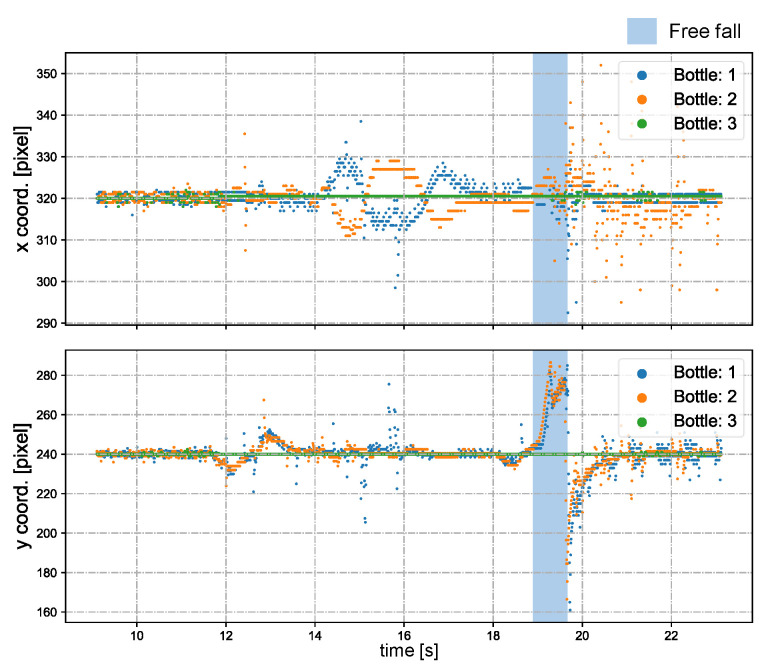
The *x* and *y* centroids of tracked bottle regions.

**Figure 14 sensors-23-04150-f014:**
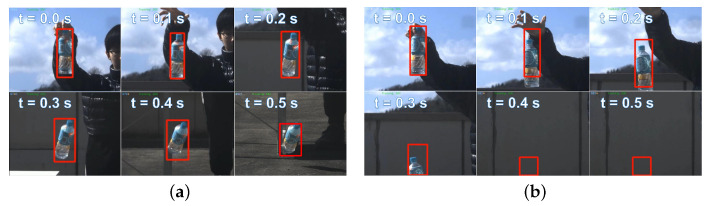
Tracking status of the free-fall of bottle 1 when tracking three bottles simultaneously: (**a**) free-fall of bottle 1 based on CNN hybrid tracking and (**b**) free-fall of bottle 1 based on YOLOv4 tracking.

**Figure 15 sensors-23-04150-f015:**
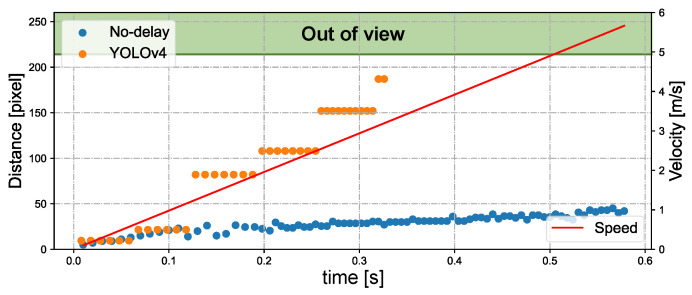
Relationship between velocity and distance from the detection ROI to the image center during free-fall bottle tracking.

**Figure 16 sensors-23-04150-f016:**
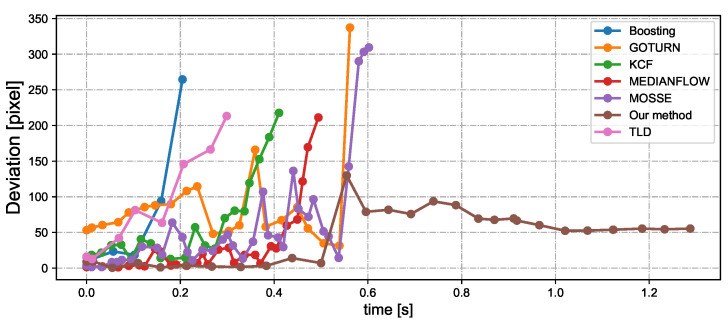
Pixel deviation value between the object position calculated by different algorithms and the object’s real position during free-falling.

**Figure 17 sensors-23-04150-f017:**
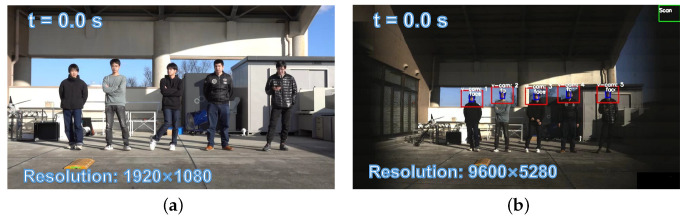
The 1920 × 1080 input images from the digital camera and panoramic stitched 9600 × 5280 images from the PTZ camera: (**a**) input image of the digital camera and (**b**) panoramic stitched image from the PTZ camera.

**Figure 18 sensors-23-04150-f018:**
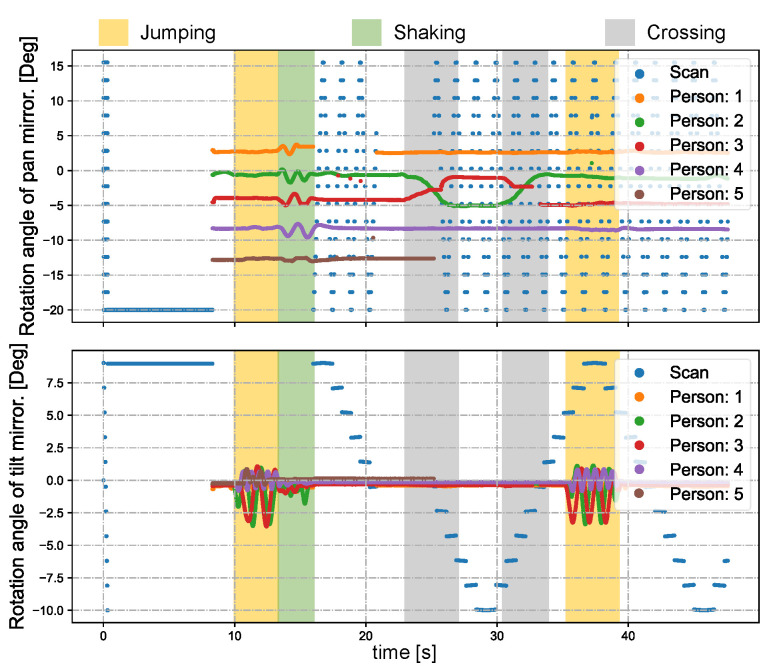
Pan and tilt angles of the galvanometer-based reflective PTZ camera when scanning and tracking multiple persons.

**Figure 19 sensors-23-04150-f019:**
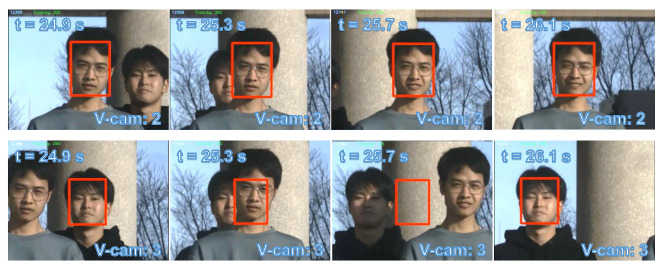
Cross-tracking of person 2 and person 3 between 24.9 and 26.1 s.

**Figure 20 sensors-23-04150-f020:**
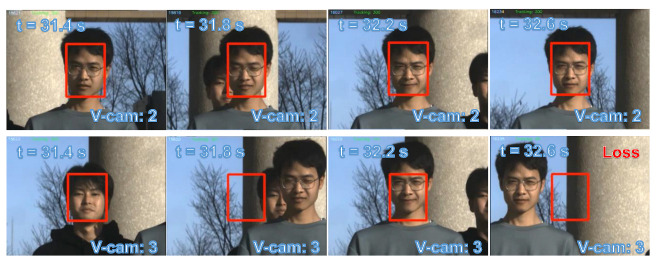
Cross-tracking of person 2 and person 3 between 31.4 and 32.6 s.

**Figure 21 sensors-23-04150-f021:**
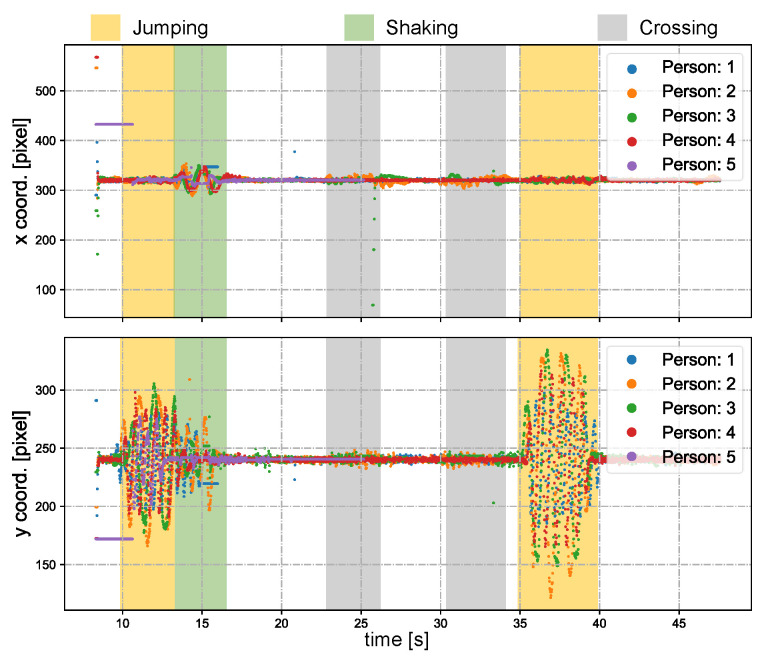
The *x* and *y* centroids of the regions with tracked people.

**Table 1 sensors-23-04150-t001:** Execution times of tracking algorithms.

	Size	64 × 64	128 × 128	256 × 256	512 × 512
Tracker					
BOOSTING		17.73	54.85	29.85	8.11
KCF		5.39	5.05	20.6	90.22
MOSSE		0.26	1.12	1.55	17.26
MIL		79.63	76.67	72.68	67.71
TLD		30.15	22.14	26.45	26.99
MEDIANFLOW		2.12	2.21	2.10	2.23
GOTURN		23.22	24.54	24.74	29.60
ours(playback)		0.27	0.41	0.77	1.82
(forward)		0.021	0.056	0.222	0.62
(YOLOv4)		33			

(unit: ms).

## Data Availability

Data sharing not applicable to this article as no datasets were generated or analysed during the current study.

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
