# Peer review of "An Active Multi-Object Ultrafast Tracking System with CNN-Based Hybrid Object Detection"

_sensors, 2023, doi:10.3390/s23084150_

Round 1

Reviewer 1 Report

The topic is very relevant and appropriate.

The abstract is lucid.

The paper proposes a visual tracking system that can detect and track multiple fast-moving appearance-varying targets simultaneously with 500 fps image processing.

The system comprises a high-speed camera and a pan-tilt galvanometer system, which can rapidly generate large-scale high-definition images of the wide monitored area using a CNN-based hybrid tracking algorithm that can track multiple high-speed moving objects simultaneously.

The novelty as presented in the paper postulates a high level of research output and rigour.

The introduction to the paper presents a very clear and concise aspect of object detection and the key research layout

The literature review on the related work is satisfactory however I noticed the citations for most of the relevant research were outdated Examples [2],  [5], [6], [7], [10], [25], [26], [28], [49], etc. This is a novel paper and needs a recent citation. In fact, the paper lacks citations of the big names in CNN, YOLO, MOT, image processing, and object detection. Citations of more than 5 years should be barely limited.

The methodology and experiments are lucid and appropriate it can however be improved.

There are a couple of spelling errors. See lines 453. 'demonstrateed by conditing'.

The discussion and conclusion are satisfactory.

Author Response

Response to Reviewer 1 Comments

Point 1: The literature review on the related work is satisfactory however I noticed the citations for most of the relevant research were outdated Examples [2],  [5], [6], [7], [10], [25], [26], [28], [49], etc. This is a novel paper and needs a recent citation. In fact, the paper lacks citations of the big names in CNN, YOLO, MOT, image processing, and object detection. Citations of more than 5 years should be barely limited.

Response 1: Thank you for your review. I have revised the paper based on your feedback. In the paper, we have referenced the latest research and notable works in the field of image processing and convolutional neural networks (CNN).

Reviewer 2 Report

This paper proposes an active multi-object ultrafast tracking system with CNN-based hybrid object detection. The HFR multiple target tracking framework designed in this paper, achieves low-latency visual feedback at hundreds of Hz, which enables the tracking of multiple fast-moving objects. However, there are still some problems to be addressed, as follows:

1) In line 64 of this paper, it is mentioned that this work achieves observation up to 20 objects at a speed of 25 fps simultaneously, but this conclusion is not verified in the experimental part. The author should prove this performance by experimental data.

2) This multi-target active tracking system proposed in this paper can only follow multiple targets moving synchronously, so when multiple targets move in the opposite direction, how the system will deal with the above problem, the author should explain.

3) In line 413 of this paper, the author mentioned they conducted experiments to track three free-falling bottles, then author should make it clear whether the pixel deviations in Figure 13 is the average value or the specific value.

4) What kind of targets can be detected by the system proposed in this paper, how does the system detect faces in the second experiment, and whether the system can give each person a specific ID number for recognition, the author should explain this problems in details.

In summary, the review opinion is major revision.

Author Response

Response to Reviewer 2 Comments

Point 1: In line 64 of this paper, it is mentioned that this work achieves observation of up to 20 objects at a speed of 25 fps simultaneously, but this conclusion is not verified in the experimental part. The author should prove this performance by experimental data.

Response 1:

Fig.1 Overview of the experimental scene.

Fig. 2 Stitching panoramic images and the position of each object in the image.

Fig. 3 HD images of 20 objects tracked simultaneously.

Thank you very much for bringing up this question. We have demonstrated the simultaneous tracking of 20 different categories of objects, including cars, bicycles, alarm clocks, and soccer balls, through a simple experiment. As shown in Figure 1, we placed 20 objects on the wall and blackboard. The objects on the wall remained stationary, while the objects on the blackboard could be made to move by moving the blackboard. Figure 2 shows the panoramic image stitched together after rapid scanning, along with the positions of the 20 different tracking targets in the panoramic image. Figure 3 presents high-definition images of the 20 targets. Since high-speed resources are divided equally, the frame rate of each target is low, and only some objects with low moving speed can be tracked. The experimental results indicate that we can track 20 different objects simultaneously at a speed of 25 frames per second.

Experiment video URL: https://youtu.be/Ko98lBIWDsk

Point 2: This multi-target active tracking system proposed in this paper can only follow multiple targets moving synchronously, so the author should explain how the system will deal with the above problem when multiple targets move in the opposite direction.

Response 2: Firstly, thank you for raising this question. Our multi-object tracking system based on a high-speed galvanometer fundamentally utilizes the principle of time-division multiplexing, by assigning continuous high-speed images to different targets, to virtualize multiple virtual cameras. As shown in the figure below, during each traversal of all targets, the galvanometer changes the camera's viewpoint at an extremely fast speed to capture images of different objects. We use the Cambridge Technology 6210H galvanometer, which can make the camera produce a 20-degree angle change in one millisecond at the fastest. Even if two objects are running in opposite directions, our galvanometer-based reflective camera can move to different positions of the objects and take pictures in one exposure time (2 ms). There are also cases where the object is very far away, and one exposure time is difficult to reach. However, our galvanometer can complete the full range of motion within two exposure times (4 ms), and we merge two frames to track distant objects. In the second experiment, there were indeed cases where different people's faces moved in different directions.

Point 3: In line 413 of this paper, the author mentioned they conducted experiments to track three free-falling bottles, then the author should make it clear whether the pixel deviations in Figure 13 are the average value or the specific value.

Response 3: Thank you for your feedback. Due to the large amount of image data generated by our high-speed visual tracking system, we encountered difficulties in saving all the images during the experiment due to limited computer memory resources. Therefore, we recorded the images at a rate of 30 frames per second. The deviation shown in Figure 13 represents the difference between the detected centre of the object using different methods and the true centre position of the object in the recorded image. It is a specific value, rather than an average value. The new point-line graph provides a more precise and visually appealing depiction of the data and enables a better understanding of the outcomes of our study.

Point 4What kind of targets can be detected by the system proposed in this paper, how does the system detect faces in the second experiment, and whether the system can give each person a specific ID number for recognition, the author should explain this problem.

Response 4: Thank you for bringing up this question. We apologize for neglecting to provide an explanation of this aspect in our paper. The detection model used in our study is a YOLOv4 model trained on the COCO dataset and fine-tuned on a facial dataset. This YOLOv4 model is capable of detecting all objects in the COCO dataset, such as footballs, water bottles, bicycles, etc. Additionally, the model can detect faces in the images. As for your question regarding whether each person can be assigned a specific ID for recognition, we utilized the Sface model to extract 128-dimensional high-dimensional features of the detected faces and utilized cosine similarity for similarity detection. We can assign a specific ID to each person for recognition. In the second experiment of our paper, there are cross-interactions among the facial targets. We continuously track the same facial target by calculating the similarity between different faces using Sface.

Reviewer 3 Report

This manuscript presents a CNN-based ultrafast multi-object detection and tracking system capable to reach a processing speed of 500 fps. Authors get high resolution images despite the large field of views.

This manuscript is well written and presented in a comprehensive manner. Although, the authors have supported their claims with extensive experiments,  I have following question:

1. How effective is the proposed visual tracking system in detecting and tracking fast-moving appearance-varrying targets simultaneously?

2. How does the system perform in crowded scenarios?

3. How does the tracking system perform in tracking moving objects  with velocities higher than 30 meters per second?

4. How does the system perform in low-light conditions?

5. How does the system handle occlusion and target crossing situations?

6. How does the system perform when tracking different types of objects?

7. How does the system compare to other tracking systems in terms of cost and hardware requirements?

I would appreciate to see answers in the revised manuscript. 

Author Response

Response to Reviewer 3 Comments

Point 1: How effective is the proposed visual tracking system in detecting and tracking fast-moving appearance-varying targets simultaneously?

Response 1: Our hybrid tracking algorithm based on CNN and template matching constantly updates the image template of the tracked object using AI with a delay of about 30 milliseconds. In the second experiment of our work, the faces of the moving persons were shaking and turning, causing significant changes in the facial image templates. However, our tracking performance was still excellent, demonstrating the effectiveness of our proposed system in detecting and tracking fast-moving targets with appearance changes.

Point 2: How does the system perform in crowded scenarios?

Response 2: In crowded scenes, there are more moving objects, which means that our system's virtual cameras will have fewer image resources allocated to each camera (resulting in decreased frame rates) when tracking more objects. The number of objects that can be tracked and their speed are contradictory: when tracking more objects, the maximum speed of trackable objects will decrease. However, as long as the AI can provide good detection results, our system can track many moving objects.

Point 3: How does the tracking system perform in tracking moving objects with velocities higher than 30 meters per second?

Response 3: The focus of this work is to propose the concept of hardware-based multi-object high-definition tracking. To test the system's limits, we determined that the maximum theoretical speed that can be tracked while simultaneously tracking three moving targets is 30 meters per second. To improve the frame rate and track faster-moving objects, we can simply reduce the number of tracked objects.

We appreciate the constructive feedback you have provided, and our next steps involve plans to address this issue. Here are some of the proposed solutions to track objects moving faster than 30 meters per second:

1)      Switch to a higher frame rate camera, such as a 1000 fps camera. With this change, the frame rate of each virtual camera will reach up to 332 frames per second while tracking three objects. With a higher frame rate, the speed of objects in continuous images decreases. Objects moving at 30 meters per second will have a motion speed between frames of less than 240 pixels.

2)      Switch to a shorter focal length lens or observe moving objects from a greater distance to increase the camera's field of view, resulting in a decrease in the objects' motion speed in pixels.

3)      Introduce motion compensation to predict the movement of objects and update control information based on the predicted position rather than the current position.

Point 4: How does the system perform in low-light conditions?

Response 4: High-speed vision systems have ultra-high frame rates, which means they have very short exposure times. For instance, a camera with 500 frames per second has an exposure time of fewer than 2 milliseconds. Consequently, almost all high-speed vision systems face difficulty operating in low-light conditions, and our system is no exception. The system is able to operate effectively in outdoor environments where the intensity of sunlight is sufficient. However, additional lighting is required to obtain higher-quality images in indoor environments.

Point 5: How does the system handle occlusion and target crossing situations?

Response 5: In the second experiment of the paper, we demonstrated the occurrence of occlusion and cross during the tracking of multiple human faces. In the case of occlusion, our virtual camera will stay at the current position for 3 seconds. If the target does not reappear after 3 seconds, it will be considered lost, and the system will restart the scanning process to register a new target. For intersection cases, multiple targets may appear in the same virtual camera. We use similarity algorithms such as ResNet and SFace to compare the similarity between multiple templates detected in the same virtual camera and the target image from the previous moment during the AI detection process. This ensures that there are no tracking errors.

Point 6: How does the system perform when tracking different types of objects?

Response 6: Our system achieves better tracking performance when tracking different types of objects, as different objects can be easily distinguished by their labels (generated by AI). Objects of the same category often share similar appearance features, making them difficult to distinguish. Due to the limitation of space, we selected representative objects of the same category for our tracking experiments.

Point 7: How does the system compare to other tracking systems in terms of cost and hardware requirements?

Response 7: To achieve the same results as our system, it would typically require the use of high-resolution, high frame-rate, wide-field-of-view panoramic cameras. Additionally, to achieve fast object tracking, there would be significant requirements for computational power on the computing platform. Both panoramic cameras and computing platforms can be expensive investments. Alternatively, multiple slower-moving pan-tilt cameras could be used to track different moving objects, which would increase hardware costs as the number of tracked objects increases.

In contrast, our system only requires a regular high-speed camera, a high-speed Galvano-mirror, and a computer with a consumer-grade configuration.

Round 2

Reviewer 2 Report

The authors have responded to the comments of the first review one by one. The supplementary experiment proves the ability of tracking 20 different kinds of objects simultaneously. After the revision, the authors have clarified the generation method of virtual camera and the algorithm used in face detection, corrected the expression of the conclusion. Generally speaking, this paper can be accepted.